# A single nucleotide polymorphism in the *HOMER1* gene is associated with sleep latency and theta power in sleep electroencephalogram

Mario Pedrazzoli[1]*, Diego Robles Mazzotti[2], Amanda Oliveira Ribeiro[1], Juliana Viana Mendes[1], Lia Rita Azeredo Bittencourt[3], Sergio Tufik[3]

1 School of Arts, Sciences and Humanities, University of São Paulo (USP), São Paulo, Brazil, 2 Center for Sleep and Circadian Neurobiology, University of Pennsylvania, Philadelphia, Pennsylvania, United States of America, 3 Department of Psychobiology, Federal University of São Paulo (UNIFESP), São Paulo, Brazil

* pedrazzo@usp.br

**Data Availability Statement:** The data presented in this manuscript was part of the EPISONO study, a large representative prospective cohort study of

## Abstract

Glutamate is the most excitatory neurotransmitter in the central nervous system and it is involved in the initiation and maintaining of waking and rapid-eye-movement (REM) sleep. Homer proteins act in the trafficking and/or clustering of metabotropic glutamate receptors, and polymorphisms in the *HOMER1* gene have been associated with phenotypes related to glutamate signaling dysregulation. In this study, we report the association of a single nucleotide polymorphism (SNP) in the *HOMER1* gene (rs3822568) with specific aspects of sleep in a sample of the Brazilian population. To accomplish this, 1,042 individuals were subjected to a full-night polysomnography, and a subset of 983 subjects had rs3822568 genotyping data available. When compared with the A allele carriers, GG genotyped individuals showed higher sleep latency, lower sleep efficiency, reduced number of arousals per hour, lower apnea-hypopnea index (AHI) and lower theta spectral power. In summary, the present findings suggest that the rs3822568 polymorphism in the *HOMER1* gene is associated with sleep EEG profiles and might have an impact on sleep quality and sleep structure, with potential to explain inter-individual variation in sleep homeostasis.

## Introduction

The sleep-wake cycle depends on a complex and well-orchestrated neuronal circuitry involving diverse neurotransmitter signaling that induce and maintain sleep and wakefulness. Glutamate, the most excitatory neurotransmitter in the central nervous system, shows dynamic changes in its levels throughout the sleep-wake states [1]. In addition, it is involved in the initiation and maintaining of waking and rapid eye movement (REM) sleep [2]. The function of glutamate as a signaling molecule in the brain is accomplished by its multiple receptor subtypes working through selective intracellular targeting mechanisms [3].

inhabitants of the city of São Paulo, Brazil. Due to restrictions from the Federal University of São Paulo Ethics Committee and Institutional Review Board (Comitê de Ética em Pesquisa # 0593/06), individual level data cannot be made available publicly. Data access can be obtained by contacting Principal Investigators of the study directly, vie e-mail (Dr. Sergio Tufik – sergio.tufik@unifesp.br; Lia Bittencourt – lia@liabittencourt.com.br). The Federal University of São Paulo Ethics Committee can be contacted at cep@unifesp.br or +55-11-5571-1062.

**Funding:** This research was supported by AFIP, FAPESP (Grant 98/14303-3) and FAPESP (Grant 11/05804-5). AFIP FAPESP (Grant 98/14303-3): - Responsible researcher: Sergio Tufik https://bv.fapesp.br/pt/auxilios/29301/center-for-sleep-studies/ FAPESP (Grant 11/05804-5): - Responsible researcher: Mario Pedrazzoli https://bv.fapesp.br/pt/auxilios/54752/caracterizacao-fenotipica-de-ritmos-circadianos-em-individuos-com-diferentes-genotipos-para-polimorf/ The funders had no role in study design, data collection and analysis, decision to publish, or preparation of the manuscript.

**Competing interests:** The authors have declared that no competing interests exist.

The Homer protein family has been implicated in the trafficking and/or clustering of metabotropic glutamate receptors, playing roles on critical glutamatergic signaling pathways [4]. These proteins are encoded by three different genes – *Homer1*, *Homer2* and *Homer3* – predominantly expressed at the nervous system as several isoforms resulting from alternative splicing events [5]. Among the splicing variants, *Homer1a* is a non-constitutive short form, induced under high neuronal activity associated with decreasing glutamate signaling [6], identified as the most specific transcriptional marker for sleep loss [7]. It was already been observed an increase in the expression of this transcript variant induced by sleep deprivation; it has been proposed that its up regulation modulates the increased neuronal glutamatergic activity found in prolonged wakefulness [7].

Single nucleotide polymorphisms (SNPs) – genomic *loci* where two or more alleles differ at a single base – have been associated with complex phenotypes, providing important molecular markers and unraveling risk *loci* for disease susceptibility [e.g. 8,9]. Polymorphisms in the *HOMER1* gene have been associated with disorders and traits related to glutamate signaling dysregulation, such as schizophrenia and cocaine dependence [10–12]. We hypothesize that, given the importance of the glutamatergic system regulating sleep and prolonged wakefulness, genetic variation in the *HOMER1* gene might explain inter-individual variability in several sleep traits measured using polysomnography. The goal of the present study was to evaluate the association of a specific SNP in the *HOMER1* gene with specific aspects of sleep in a representative sample of the Brazilian population.

## Material and methods

### Subjects

Subjects consisted of a sample of 1,042 individuals from the São Paulo Epidemiological Sleep Study (EPISONO), a population-based survey to represent São Paulo city population, delineating the epidemiological profile of sleep disorders in a metropolitan Brazilian city according to gender, age, and socioeconomic status in 2007. All individuals answered questionnaires about socioeconomic, demographic, lifestyle and general health factors. Blood samples were collected to investigate genetic traits and polysomnographic recordings were taken to access objective sleep quality. This study was approved by the Ethics Committee of the Federal University of São Paulo (CEP 0593/06) and was registered under ClinicalTrials.gov (NCT00596713; Epidemiology of sleep disturbances among adult population of the São Paulo City). All participants signed up informed consent forms.

### Polysomnography and clinical assessment

A full-night polysomnography was performed using a digital system (EMBLA® S7000, Embla Systems, Inc., Broomfield, CO, USA) at the sleep laboratory. Physiological variables were monitored continuously, and recordings were scored according to standardized criteria [13,14]. Obstructive sleep apnea syndrome (OSAS) was considered positive if individuals had an apnea-hypopnea index (AHI) between 5 and 14.9 and presented at least one of the following complaints: loud snoring, daytime sleepiness, fatigue, and breathing interruptions during sleep. Subjects with an AHI $\geq$15 were also considered positive, regardless whether they presented complaints [15].

### Spectral analysis of sleep EEG

A specific syntax in R (version 2.10.1) was used for the spectral analysis of the sleep electroencephalography (EEG), performed according to previously published studies [16,17]. Briefly describing, waves from C3-A2, C4-A1, O1-A2, and O2-A1 derivations were decomposed into

delta (<4 Hz), theta (4–7.9 Hz), alpha 1 (8–9.9 Hz), alpha 2 (10–12.9 Hz), beta 1 (13–17.9 Hz), beta 2 (18–29.9 Hz), and gamma (≥30 Hz) frequency bands using fast Fourier transformation, with a sampling rate of 200 Hz, using epochs of 20 seconds. The filter settings used were in accordance to standard criteria of sleep EEG data acquisition (low frequency filter = 0.3 Hz; high frequency filter = 35 Hz; time constant = 0.3 seconds; and notch filter = 60 Hz). Artifact removal was performed as previously described [16].

### rs3822568 genotyping

Genomic DNA was extracted from the volunteers' white blood cells through the salting out of the cellular proteins and precipitation with a saturated NaCl solution [18]. *HOMER1* SNP rs3822568 was amplified by PCR (Polymerase Chain Reaction) using the primers H1F: 5′CCTGTTCACTGAGAAGAGCCTA3′ and H1R: 3′GAAATACAGCAGCCCGTCAT5′, under the following thermal conditions: 5 min at 95˚C; 35 cycles of 0.5 min at 95˚C, 0.5 min at 66˚C, and 0.5 min at 72˚C; 5 min at 72˚C; held at 4˚C. The 25 μl PCR mixes included 2.5 μl 10X PCR buffer, 0.75 μl $MgCl_2$ (1.5 mM), 0.5 μl of each primer, 0.5 μl of dNTP mix (2mM dATP, 2mM dCTP, 2mMdTTP, and 2mM dGTP), 0.25 μl of Taq DNA Polymerase, and 1.0 μl of DNA template (100ng/μl). Due to the presence of restriction sites inside the *HOMER1* sequence, genotypes were determined using the Restriction Fragment Length Polymorphism (RFLP) method. Restriction endonuclease digestions were carried out directly on PCR products in 20 μl reactions containing 0.2 μl 10X buffer solution, 0.25 μl *PvuII* (New England Biolabs), and 5.0 μl PCR product. The *PvuII* enzyme cleaves the palindromic sequence CAG^CTG, providing a three-band profile (397pb, 218pb, and 178pb) for subjects carrying the A/G genotype; a two-band profile (218pb and 178pb) for homozyguous A/A; and the single-band profile (397pb) for homozyguous G/G. Digestion took place at 37˚C for 16 hours, and the obtained profiles were visualized in 1% agarose gels, stained with ethidium bromide.

### Statistical analysis

The chi-square test was used to verify whether *HOMER1* genotype frequencies were distributed according to the Hardy-Weinberg Equilibrium. One-way Analysis of Variance (ANOVA) followed by Bonferroni *post hoc* test was used to verify the effect of *HOMER1* polymorphism on the z-score standardized polysomnographic parameters and sleep EEG spectral data. Also, the chi-square test was performed to compare genotype frequencies between individuals with and without OSAS. Furthermore, a set of 31 ancestry informative markers was used to estimate the genetic ancestry proportions of the population as previously described [19]. General linear models (GLM) were applied to verify the effect of potential confounders (age, sex, body mass index and ancestry proportions) on the genetic association results of the continuous variables. Tests were performed using PAWS 18.0 (SPSS, Inc.), with a significance level of 0.05. Results are represented as mean [standard deviation].

## Results

A total of 983 individuals (42.43 [14.36] years; 55.8% females) had valid genotypes for the *HOMER1* polymorphism rs3822568. Genotype frequencies for this SNP were 219 (22.3%), 469 (47.7%) and 295 (30.0%) for AA, AG and GG, respectively, and genotype distribution did not show deviations from Hardy-Weinberg Equilibrium ($\chi^2$ = 1.58; p = 0.209).

We verified the association of *HOMER1* rs3822568 genotypes with a number of polysomnographic parameters, as show in Table 1. We found significant associations between rs3822568 and sleep latency ($F_{(2,980)}$ = 6.389, p = 0.002), sleep efficiency ($F_{(2,980)}$ = 2.962, p = 0.033), number of arousals per hour ($F_{(2,980)}$ = 6.075, p = 0.002) and apnea-hypopnea

**Table 1. Sleep parameters measured by polysomnography comparing the three *HOMER1* rs3822568 SNP genotypes.**

| Polysomnographic parameter | rs3822568 genotype | N | Mean (SD) | $F_{(2,980)}$ | p |
|---|---|---|---|---|---|
| Sleep latency (minutes) | AA | 219 | 17.14 (21.82) | 6.389 | 0.002 |
| | AG | 469 | 14.57 (16.23) | | |
| | GG | 295 | 20.48 (29.74) * | | |
| REM sleep latency (minutes) | AA | 219 | 98.71 (54.99) | 0.596 | 0.551 |
| | AG | 469 | 102.03 (54.15) | | |
| | GG | 295 | 98.01 (52.22) | | |
| Sleep total time (minutes) | AA | 219 | 346.19 (85.67) | 2.962 | 0.052 |
| | AG | 469 | 347.28 (71.43) | | |
| | GG | 295 | 333.82 (80.64) | | |
| Sleep efficiency (%) | AA | 219 | 81.18 (13.9) | 3.433 | 0.033 |
| | AG | 469 | 82.94 (11.71) | | |
| | GG | 295 | 80.56 (13.95) * | | |
| Stage 1 (%) | AA | 219 | 4.86 (3.72) | 1.725 | 0.179 |
| | AG | 469 | 4.68 (3.28) | | |
| | GG | 295 | 4.33 (3.19) | | |
| Stage 2 (%) | AA | 219 | 54.87 (8.75) | 0.284 | 0.753 |
| | AG | 469 | 54.67 (9.57) | | |
| | GG | 295 | 54.28 (9.23) | | |
| Stages N3 (%) | AA | 219 | 21.23 (7.35) | 3.276 | 0.038 |
| | AG | 469 | 21.53 (8.23) | | |
| | GG | 295 | 22.85 (8.29) | | |
| REM sleep (%) | AA | 219 | 19.04 (6.72) | 0.771 | 0.463 |
| | AG | 469 | 19.13 (6.48) | | |
| | GG | 295 | 18.54 (6.47) | | |
| Awake time (minutes) | AA | 219 | 62.72 (50.2) | 0.868 | 0.420 |
| | AG | 469 | 57.83 (44.73) | | |
| | GG | 295 | 59.74 (43.19) | | |
| Number of arousals per hour | AA | 219 | 16.61 (11.6) | 6.075 | 0.002 |
| | AG | 469 | 15.16 (11.69) | | |
| | GG | 295 | 13.27 (9.12) * | | |
| Apnea-hypopnea index | AA | 219 | 9.89 (14.98) | 3.988 | 0.019 |
| | AG | 469 | 8.33 (13.18) | | |
| | GG | 295 | 6.63 (11.13) * | | |

One-way ANOVA on z-score standardized measurements; SD: standard deviation; REM: rapid eye-movement.

* Indicates p<0.05 when compared to the other genotypes inside each variable.

index ($F_{(2,980)}$ = 3.988, p = 0.019). In summary, GG genotype carriers showed higher sleep latency in minutes (20.48 [29.74]) than AG genotype carriers (14.57 [16.23], Bonferroni *post hoc* p = 0.001) and lower sleep efficiency (80.56 [13.95]) than AG genotype carriers (82.94 [11.71], Bonferroni *post hoc* p = 0;040). In addition, GG carriers showed lower arousals per hour index (13.27 [9.12]) and AHI (6.63 [11.13]) than AA genotype carriers (16.61 [11.6]; Bonferroni *post hoc* p = 0.002; and 9.89 [14.98]; Bonferroni *post hoc* p = 0.016, respectively). We also found a trend for association between *HOMER1* rs3822568 polymorphism and the percentage of stages N3, but no significant associations were found after the *post hoc* test (Table 1). Also, despite the association between this SNP and the apnea-hypopnea index levels, no significant association with the presence of OSAS was found ($\chi^2$ = 4.104; p = 0.128).

**Table 2. Significant associations between *HOMER1* rs3822568 polymorphism genotypes and theta (4–7.9 Hz) spectral power in sleep electroencephalogram.** Results are shown by derivation and sleep stage.

| Derivation | Sleep Stage | rs3822568 Genotype | | | | | | | | | df | F | p-value | Adjusted p-value |
|---|---|---|---|---|---|---|---|---|---|---|---|---|---|---|
| | | AA | | | AG | | | GG | | | | | | |
| | | N | Mean | SD | N | Mean | SD | N | Mean | SD | | | | |
| O2-A1 | 1 | 173 | 4.34 | 1.25 | 351 | 4.49 | 1.23 | 231 | 4.19 | 1.15 | 2,752 | 4.412 | 0.012 | 0.035 [a] |
| | 2 | 188 | 4.24 | 1.18 | 384 | 4.42 | 1.24 | 245 | 4.17 | 1.16 | 2,814 | 3.679 | 0.026 | 0.086 |
| | REM | 187 | 4.09 | 1.20 | 382 | 4.15 | 1.10 | 243 | 3.87 | 1.08 | 2,809 | 4.847 | 0.008 | 0.029 [b] |
| O1-A2 | 1 | 173 | 4.48 | 1.14 | 351 | 4.52 | 1.21 | 231 | 4.25 | 1.19 | 2,752 | 3.769 | 0.024 | 0.062 |
| | REM | 187 | 4.29 | 1.15 | 382 | 4.24 | 1.09 | 243 | 3.97 | 1.10 | 2,809 | 5.59 | 0.004 | 0.008 [c] |
| C3-A2 | 1 | 173 | 3.92 | 1.17 | 351 | 3.95 | 1.11 | 231 | 3.70 | 1.03 | 2,752 | 3.925 | 0.020 | 0.089 |
| | 2 | 188 | 3.30 | 1.03 | 384 | 3.40 | 1.04 | 245 | 3.16 | 0.94 | 2,814 | 4.4 | 0.013 | 0.074 |
| | REM | 187 | 4.00 | 1.06 | 382 | 4.02 | 1.03 | 243 | 3.80 | 1.02 | 2,809 | 3.826 | 0.022 | 0.077 |
| C4-A1 | REM | 187 | 3.84 | 1.01 | 382 | 3.86 | 1.04 | 243 | 3.63 | 1.00 | 2.809 | 4.09 | 0.017 | 0.073 |

One-Way ANOVA on z-score standardized measurements; SD: standard deviation; df: degrees of freedom (between groups, within groups); Adjusted p-value by sex, age, BMI and European ancestry proportion in general linear models.

[a] Bonferroni *post hoc* test for AG x GG: p = 0.041

[b] Bonferroni *post hoc* test for AG x GG: p = 0.023

[c] Bonferroni *post hoc* test for AA x GG: p = 0.018; for AG x GG: p = 0.020

To verify the effect of potential confounders on the identified associations, we fitted GLMs using *HOMER1* rs3822568 genotypes as well as sex, age, BMI and European ancestry proportion derived from ancestry informative markers as independent variables and the z-score of each significantly associated parameter in univariate analyses as dependent variables in each model. After adjustment for the studied confounders, *HOMER1* rs3822568 polymorphism was significantly and independently associated only with sleep latency, regardless of other variables in the model (p = 0.005).

Comparing the spectral power of each studied bandwidth in each sleep stage for each EEG derivation, we found significant associations between *HOMER1* rs3822568 polymorphism and theta spectral power in all EEG derivations, even after adjustment for sex, age, BMI and European ancestry proportion. Overall, GG genotype carriers showed lower theta spectral power in stage 1, stage 2 and REM sleep; however, after adjustment for potential confounders, associations remained significant only in the occipital derivations and in sleep stage 1 and REM sleep (Table 2). No significant associations with other bandwidths were found.

## Discussion

Several previous studies revealed the interaction of Homer proteins with metabotropic glutamate receptors and the potential role of these proteins in the trafficking and/or clustering of the receptors in various cell types [4,20–23]. Studies interfering with the normal expression of *HOMER* genes suggest the involvement of these gene products in animal behavior, from *Drosophila* to mammals [5]. Also, genetic variation in *HOMER1* seem to be associated with drug dependency and abuse [11] and mental disease [10].

In the present study, we report the association between rs3822568 in the *HOMER1* gene with sleep latency, sleep efficiency, number of arousals per hour, AHI and theta spectral power in healthy subjects. The analyzed SNP consists of a genetic variant found at the 3' untranslated region (3'UTR) of the *HOMER1* gene, for which the ancestral allele is a guanine (G) and the alternative allele is an adenine (A).

Here, we found that homozygous individuals for the G allele, on average, experience higher sleep latency times, lower sleep efficiency, less arousals per hour and lower AHI than subjects carrying the A allele. This suggests that the presence of the G allele is associated with changes in sleep onset – leading to the higher latency times – but not the sleep maintenance – since no associations with arousals per hour were found. In this sense, the observed lower sleep efficiency might be attributed to the long sleep latency rather than to a higher sleep fragmentation caused by many arousals during the sleep time [24,25].

On the other hand, even if AHI is a common measure for the diagnosis and severity determination of OSAS, rs3822568 did not show any association with the presence of the syndrome according to the definitions we used [15]. This corroborates that other factors besides the genetic background might influence the manifestation of the obstructive sleep apnea syndrome, such as environmental and developmental factors [26]. Another aspect that needs to be taken in consideration is that the definition of hypopnea is not consensual among sleep researches [27], and some studies suggest than an AHI cutoff of 5 is too low, especially for elderly people [28]. Moreover, we also observed that gender, age, BMI and genetic ancestry might influence the association between the investigated polymorphism and sleep traits, except sleep latency.

It is known that oscillations in the sleep EEG reflect the homeostatic regulation of sleep [29]. Common variation in genes involved in sleep homeostasis has been associated with sleep-related traits. Mazzotti and colleagues [16] found an association between a SNP in the adenosine deaminase gene and higher delta and theta EEG spectral powers, indicating a higher sleep intensity in carriers of the alternative allele. In our investigation, we observed that GG genotype carriers showed, on average, lower theta spectral power in stages 1, 2 and REM sleep – the associations remained significant only in the occipital derivations and in stage 1 and REM sleep after the adjustment for potential confounders (Table 2). These findings add on the results regarding the association between GG genotype and lower sleep efficiency, and suggests that changes in EEG power associated with this SNP might explain the differences found in sleep efficiency.

Naidoo and colleagues [30] demonstrated *Homer1* upregulation during wakefulness and its down regulation at sleep is not a simple correlate, but a matter of cause and effect. Using *Drosophila* and mouse models, these authors showed that *Homer1a/homer1a* knockout leads to reduced and fragmented sleep in flies, while it causes inability to sustain the wake state in the rodents. Thus, it is possible that genetic variation in the *Homer* genes could modulate the subtle balance of HOMER proteins in the regulation of glutamatergic neurotransmission and yield different states of neuronal excitability in the nervous system, affecting sleep.

Polymorphisms in the *Homer* genes could, therefore, modulate the interactions among HOMER proteins in regulating glutamatergic neurotransmission and yield different states of neuronal excitability in the nervous system affecting sleep state. Considering that the 3'UTR is a key post-transcriptional regulation site, figuring a target for non-coding RNA based regulatory mechanisms such as microRNAs [31], polymorphisms at the 3'UTR may lead to a differential regulation of gene expression [32]. This could explain the functional consequences of the studied SNP on variability of sleep-related traits presented in this study.

In the view of recent studies about glutamatergic neurotransmission, sleep homeostasis and synaptic plasticity, *Homer1* is strictly linked to the (mGluR5) function. It selectively uncouples mGluR5 from the effector targets in the postsynaptic density [33] that results in attenuation of mGluR5-mediated activation of phospholipase C and downstream signaling [34]. In humans, increased mGluR5 availability after sleep deprivation was tightly associated with increased propensity to fall asleep [35] and considering the interplay between *Homer1* and mGluR5, it has been suggested that *Homer1a* serves as a molecular integrator of arousal and sleep need [36].

In that way, we may cautionary propose the SNP described here could mediates the integration *Homer1*/mGluR5 through 3'UTR mechanisms cited above, and therefore would mediates a compensatory mechanism to promote wakefulness in the sleep-deprived state like insomnia states.

When considering that the relationship between disease susceptibility and genetic variation is complex, and for the most phenotypes more than one gene is generally involved, care has to be taken when considering just one SNP in one gene to draw conclusions about a complex and multifactorial phenotype as sleep. Nevertheless, when considering the physiological mechanisms related to glutamate neurotransmission and previous associations with sleep, *HOMER1* sounds like a good candidate gene for sleep phenotypes related to extended wake like insomnia.

*HOMER1* has many single nucleotide polymorphisms along its sequence. According to the 1000 Genomes Project data, in American populations rs3822568 is in strong linkage disequilibrium with at least five other SNPs (**S1 Fig**), although all of them are localized in intergenic regions and therefore have low potential to yield genetic consequences. Also, neither of them has been associated with any relevant clinical phenotype. This fact does not exclude the possibility of other SNPs and even phased haplotypes play important roles in the genetic control of the sleep parameters analyzed in this study.

In this context, the present findings suggest that the rs3822568 polymorphism in the *HOMER1* gene is associated with impacts on sleep quality and sleep structure, as evidenced by higher sleep latency and lower EEG theta power in individuals carrying the GG genotype. Thus, this SNP may be an important source of variation in sleep homeostasis, probably due to the regulation of the glutamatergic signaling pathways.

## Supporting information

**S1 Fig. Haplotype and linkage disequilibrium (LD) map.** S1A Fig LD plot of six SNPs in *HOMER1* gene. Dark red shading denotes D' values (i.e., 99 means D' of 0.99). Squares with no numbers indicates D' of 1. The numbers 1, 13, 22, 26, 55 and 67 in the first line indicate the position of the six strongest (D' ≥ 0.90) SNP linkage among 67 SNPs in the region. S1B Fig Inferred haplotypes for the LD block with frequencies). HaploView (version 4.2).
(TIF)

## Acknowledgments

We thank Luiz Otávio Bastos Esteves and Guilherme Silva Umemura for their assistance and comments that greatly improved the manuscript.

## Author Contributions

**Conceptualization:** Mario Pedrazzoli, Diego Robles Mazzotti, Amanda Oliveira Ribeiro, Juliana Viana Mendes, Lia Rita Azeredo Bittencourt.

**Formal analysis:** Mario Pedrazzoli, Diego Robles Mazzotti, Amanda Oliveira Ribeiro, Lia Rita Azeredo Bittencourt.

**Funding acquisition:** Sergio Tufik.

**Investigation:** Mario Pedrazzoli, Amanda Oliveira Ribeiro, Lia Rita Azeredo Bittencourt.

**Methodology:** Mario Pedrazzoli, Diego Robles Mazzotti, Amanda Oliveira Ribeiro, Lia Rita Azeredo Bittencourt.

**Project administration:** Sergio Tufik.

**Resources:** Mario Pedrazzoli.

**Supervision:** Mario Pedrazzoli, Diego Robles Mazzotti, Lia Rita Azeredo Bittencourt.

**Writing – original draft:** Mario Pedrazzoli, Diego Robles Mazzotti, Amanda Oliveira Ribeiro, Lia Rita Azeredo Bittencourt.

**Writing – review & editing:** Mario Pedrazzoli, Diego Robles Mazzotti, Juliana Viana Mendes, Lia Rita Azeredo Bittencourt.

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
