## [Decision Letter · Decision Letter 0]

3 Mar 2020

PONE-D-19-26778

A single nucleotide polymorphism in the HOMER1 gene is associated with sleep latency and theta power in sleep electroencephalogram

PLOS ONE

Dear Prof Pedrazzoli,

Thank you for submitting your manuscript to PLOS ONE. After careful consideration, we feel that it has merit but does not fully meet PLOS ONE’s publication criteria as it currently stands. Therefore, we invite you to submit a revised version of the manuscript that addresses the points raised during the review process.

ACADEMIC EDITOR: Please consider the minor concerns highlighted by both reviewers: 

1) Better discuss that the study of only one gene (and one SNP) and the multifactoral phenotypes with possible multiple genetic factors 

We would appreciate receiving your revised manuscript by Apr 17 2020 11:59PM. To enhance the reproducibility of your results, we recommend that if applicable you deposit your laboratory protocols in protocols.io, where a protocol can be assigned its own identifier (DOI) such that it can be cited independently in the future. For instructions see: http://journals.plos.org/plosone/s/submission-guidelines#loc-laboratory-protocols

We look forward to receiving your revised manuscript.

Kind regards,

Andrea Romigi, M.D., Ph.D

Academic Editor

PLOS ONE

Journal Requirements:

Reviewers' comments:

Reviewer's Responses to Questions

**Comments to the Author**

1. Is the manuscript technically sound, and do the data support the conclusions?

Reviewer #1: Partly

Reviewer #2: Yes

2. Has the statistical analysis been performed appropriately and rigorously? 

Reviewer #1: Yes

Reviewer #2: Yes

3. Have the authors made all data underlying the findings in their manuscript fully available?

Reviewer #1: Yes

Reviewer #2: Yes

4. Is the manuscript presented in an intelligible fashion and written in standard English?

Reviewer #1: Yes

Reviewer #2: Yes

5. Review Comments to the Author

Reviewer #1: In this study, the authors evaluated the association of a specific SNP in the HOMER1 gene with specific aspects of sleep in a representative sample of the Brazilian population. They in fact hypothesize that, given the importance of the glutamatergic system regulating sleep and prolonged wakefulness, genetic variation in the HOMER1 gene might explain inter-individual variability in several sleep traits measured using polysomnography.

Even if they studied only a gene and only one SNP of this gene, the study is well conducted and the sample size of studied subjects is consistent

The research is quite original, the experiments and the statistical analyses were conducted in a proper way and the paper is written in a clear and intelligible way.

Some minor issues are to be considered:

The author should better discuss that this study involve only one gene (and one SNP) but the phenotypes they are discussing is multifactorial and the genetic factors involved can be numerous. The gene they are studying may give a contribution but it does not explain the whole genetic variability of these phenotypes. They should also discuss about the possibility of other SNPs (and HAPLOTYPES) of this gene involved Are already known other SNPs? Could they sequence the entire gene in a group of subjects in order to study this variability and the existence of other SNPs? They should at least discuss it

They should also discuss about the functional role and meaning of the SNP they analyzed

Reviewer #2: In the current manuscript Pedrazzoli et al., examine the association between overnight sleep parameters and a polymorphism in the Homer1 gene in a Brazilian population. The authors report significant genotype interactions in the Homer1 3'UTR and sleep latency, efficiency and other parameters, as well as an interaction with Theta power in the EEG recordings. Recent studies have revealed genetic links with the Homer1 gene and psychiatric conditions. Basic research has also shown an important role for the Homer1 gene in sleep behavior in rodents and insects. The current work is presented in a straight forward manner and the data collection has been done rigorously and appropriately.

Considering the material covered in this manuscript it would be highly appropriate to include in the bibliography a recent high profile paper on the role of Homer1a in synaptic plasticity during sleep. (Diering et al., 2017 Science, PMID: 28154077). This paper shows that Homer1a protein is important to regulate mGluR1/5 signaling during sleep and for sleep-dependent remodeling of synapses. A second paper Holst et al., 2017 eLife, PMID: 28980941 shows that mGluR5 is upregulated during sleep deprivation in humans and is required for appropriate response to sleep deprivation in mice.

I have no other concerns about the current manuscript.

6. PLOS authors have the option to publish the peer review history of their article (what does this mean?). If published, this will include your full peer review and any attached files.

Reviewer #1: No

Reviewer #2: Yes: Graham Hugh Diering

---

## [Author Response · Author response to Decision Letter 0]

17 Apr 2020

# EDITOR: To enhance the reproducibility of your results, we recommend that if applicable you deposit your laboratory protocols in protocols.io, where a protocol can be assigned its own identifier (DOI) such that it can be cited independently in the future. For instructions see: http://journals.plos.org/plosone/s/submission-guidelines#loc-laboratory-protocols

RESPONSE: The majority of our protocols involve standard methods such as the scores of polysomnography data and SNP genotyping. We have specified the thermal conditions in Material and Methods section (“rs3822568 Genotyping” subsection). We do not think it is necessary to specify other conventional protocols involved in our study.

# REVIEWER 1: The author should better discuss that this study involve only one gene (and one SNP) but the phenotypes they are discussing is multifactorial and the genetic factors involved can be numerous. The gene they are studying may give a contribution but it does not explain the whole genetic variability of these phenotypes.

RESPONSE: Thank you for pointing it out. We have added a paragraph (L229-234) in Discussion section arguing about this matter:

“When considering that the relationship between disease susceptibility and genetic variation is complex, and for the most phenotypes more than one gene is generally involved, care has to be taken when considering just one SNP in one gene to draw conclusions about a complex and multifactorial phenotype as sleep. Nevertheless, when considering the physiological mechanisms related to glutamate neurotransmission and previous associations with sleep, HOMER1 sounds like a good candidate gene for sleep phenotypes related to extended wake like insomnia.”

# REVIEWER 1: They should also discuss about the possibility of other SNPs (and HAPLOTYPES) of this gene involved Are already known other SNPs? Could they sequence the entire gene in a group of subjects in order to study this variability and the existence of other SNPs? They should at least discuss it.

RESPONSE: We appreciate your suggestion. However, it is not possible at this time to sequence the samples. Therefore, we have added a supplemental material informing about the linkage disequilibrium (S1A Figure) and also a paragraph (L235-240) discussing the possibilities for other SNPs in HOMER1 gene that might have a role in the phenotype:

“HOMER1 has many single nucleotide polymorphisms along its sequence. According to the 1000 Genomes Project data, in American populations rs3822568 is in strong linkage disequilibrium with at least five other SNPs (S1 Figure), although all of them are localized in intergenic regions and therefore have low potential to yield genetic consequences. Also, neither of them has been associated with any relevant clinical phenotype. This fact does not exclude the possibility of other SNPs and even phased haplotypes play important roles in the genetic control of the sleep parameters analyzed in this study.”

# REVIEWER 1: They should also discuss about the functional role and meaning of the SNP they analyzed.

RESPONSE: Great point. We have added a new paragraph (L220-L228) in Discussion, which we cautionary suggest (L226-228) the role of rs3822568 in a physiological context:

“In the view of recent studies about glutamatergic neurotransmission, sleep homeostasis and synaptic plasticity, Homer1 is strictly linked to the (mGluR5) function. It selectively uncouples mGluR5 from the effector targets in the postsynaptic density (33) that results in attenuation of mGluR5-mediated activation of phospholipase C and downstream signaling (34). In humans, increased mGluR5 availability after sleep deprivation was tightly associated with increased propensity to fall asleep (35) and considering the interplay between Homer1 and mGluR5, it has been suggested that Homer1a serves as a molecular integrator of arousal and sleep need (36). In that way, we may cautionary propose the SNP described here could mediates the integration Homer1/mGluR5 through 3’UTR mechanisms cited above, and therefore would mediates a compensatory mechanism to promote wakefulness in the sleep-deprived state like insomnia states.”

#REVIEWER 2: In the current manuscript Pedrazzoli et al., examine the association between overnight sleep parameters and a polymorphism in the Homer1 gene in a Brazilian population. The authors report significant genotype interactions in the Homer1 3'UTR and sleep latency, efficiency and other parameters, as well as an interaction with Theta power in the EEG recordings. Recent studies have revealed genetic links with the Homer1 gene and psychiatric conditions. Basic research has also shown an important role for the Homer1 gene in sleep behavior in rodents and insects. The current work is presented in a straight forward manner and the data collection has been done rigorously and appropriately.

Considering the material covered in this manuscript it would be highly appropriate to include in the bibliography a recent high profile paper on the role of Homer1a in synaptic plasticity during sleep. (Diering et al., 2017 Science, PMID: 28154077). This paper shows that Homer1a protein is important to regulate mGluR1/5 signaling during sleep and for sleep-dependent remodeling of synapses. A second paper Holst et al., 2017 eLife, PMID: 28980941 shows that mGluR5 is upregulated during sleep deprivation in humans and is required for appropriate response to sleep deprivation in mice.

I have no other concerns about the current manuscript.

RESPONSE: Thank you for the suggestion. We have added a new paragraph (L220-L228) in Discussion with references suggested above and other:

“In the view of recent studies about glutamatergic neurotransmission, sleep homeostasis and synaptic plasticity, Homer1 is strictly linked to the (mGluR5) function. It selectively uncouples mGluR5 from the effector targets in the postsynaptic density (33) that results in attenuation of mGluR5-mediated activation of phospholipase C and downstream signaling (34). In humans, increased mGluR5 availability after sleep deprivation was tightly associated with increased propensity to fall asleep (35) and considering the interplay between Homer1 and mGluR5, it has been suggested that Homer1a serves as a molecular integrator of arousal and sleep need (36). In that way, we may cautionary propose the SNP described here could mediates the integration Homer1/mGluR5 through 3’UTR mechanisms cited above, and therefore would mediates a compensatory mechanism to promote wakefulness in the sleep-deprived state like insomnia states.”

---

## [Editor Report · Decision Letter 1]

17 Jun 2020

A single nucleotide polymorphism in the HOMER1 gene is associated with sleep latency and theta power in sleep electroencephalogram

PONE-D-19-26778R1

Dear Dr. Pedrazzoli,

We’re pleased to inform you that your manuscript has been judged scientifically suitable for publication and will be formally accepted for publication once it meets all outstanding technical requirements.

Kind regards,

Andrea Romigi, M.D., Ph.D

Academic Editor

PLOS ONE
---

## [Editor Report · Acceptance letter]

26 Jun 2020

PONE-D-19-26778R1 

A single nucleotide polymorphism in the *HOMER1* gene is associated with sleep latency and theta power in sleep electroencephalogram 

Dear Dr. Pedrazzoli:

I'm pleased to inform you that your manuscript has been deemed suitable for publication in PLOS ONE. Congratulations! Your manuscript is now with our production department. 

Kind regards, 

on behalf of

Dr. Andrea Romigi 

Academic Editor

PLOS ONE